# Changes in instrumental activities daily living limitations and their associated factors according to gender in community-residing older adults: A longitudinal cohort study

**SeolHwa Moon[1], Eunmi Oh[2], Daum Chung[3], Gwi-Ryung Son Hong[3]***

**1** Department of Nursing, Hoseo University, Cheonan, Republic of Korea, **2** Research Institute of Nursing Science, Hanyang University, Seoul, Republic of Korea, **3** College of Nursing, Hanyang University, Seoul, Republic of Korea

* grson@hanyang.ac.kr

## Abstract

### Background

Increases in dependence among older adults cause increases in care needs and social care burden. Instrumental activities of daily living (IADL) are often used to assess the independence of older adults residing in communities. Various factors affect IADL limitations, but few studies confirm gender differences in IADL limitations in older adults. This study explored the changes in incidence of IADL limitations across 14 years and identified the factors associated with IADL limitations according to gender among older adults in Korea.

### Method

This study uses secondary data analysis with 2006–2020 data from the Korean Longitudinal Study of Aging (KLoSA), a longitudinal cohort study. Among the total 10,254 participants, 1,230 adults aged 65 years and over who met the criteria were included in the final analysis. For each IADL item, a limitation was defined when the response was partial or complete dependence. Multivariate logistic regression was performed to identify the factors (in 2006) associated with IADL limitations in 2020.

### Results

The mean age of the participants at baseline was 69.64 years (SD 3.93), and 61.0% of participants were female. Total scores for IADL limitations increased gradually across 14 years in all participants, and observed changes were statistically significant. The top three ranked items of IADL limitations differed according to gender in 2020: the top limitations in men were preparing meals, laundry, and using public transportation, and the top limitations in women were using public transportation, going out, and handling money. Factors associated with total IADL limitations were grip strength in men and age, dementia, fear of fall, and grip strength in women. Factors associated with the top three ranked items of IADL limitations differed according to gender.

**Data Availability Statement:** The present study was used KLoSA data conducted by the Korea Employment Information Service and is publicly

accessible at https://survey.keis.or.kr/eng/klosa/databoard/List.jsp.

**Funding:** This research was supported by the Translational Research Program for Care Robots funded by the Ministry of Health & Welfare, Republic of Korea (grant number: HK21C0008, PI, Gwi-Ryung Son Hong). There was no additional internal and external funding received for this study. The funders had no role in study design, data collection and analysis, decision to publish, or preparation of the manuscript.

**Competing interests:** The authors have declared that no competing interests exist.

## Conclusion

The incidence of IADL limitations gradually increased in all participants over a 14-year period. In older adults in Korea, gender differences were confirmed in the factors associated with IADL limitations, as well as in the main limited activities. To minimize IADL limitations in community-residing older adults, it is necessary to plan tailored interventions.

## Introduction

As phenomena of general well-being, activities of daily living (ADL) include the ability to independently perform daily activities to sustain personal and social life [1]. ADL indexes are divided into basic ADL and instrumental activities of daily living (IADL), and these tools are widely used as indicators for assessing the functional status of older adults. Basic ADL include activities for survival, such as basic function, self-care, and mobility, while IADL include activities for more complex tasks required to live in communities, such as money management, housework, and use of public transportation [2]. Recognized as hierarchically superior for more complex tasks than ADL, limitations in IADL among older adults are generally experienced before ADL limitations [3].

Older populations experience some level of difficulties in performing IADL with deterioration of physical and cognitive functioning [4]. Increased dependence in older adults is associated with hospitalization and admission to long-term care facilities, leading to increase in morbidity and mortality from diseases along with a decrease in quality of life [5, 6]. Increased dependence in older adults also increases social burden due to increased care needs and medical costs [7]. To reduce the social burden caused by functional limitations of older adults, it is necessary to prevent temporary limitations and the associated need for care.

Limitations in IADL in older adults increase with age, with previous research reporting gradual increase over a 10-year period [4], as well as in a study with just three years of follow-up [8]. Most previous research examines changes in the incidence of IADL limitations in all segments of older populations together. Because incidence and limitations in IADL vary based on gender for different activities, it is important to assess these changes by gender to understand functional limitations among older adults.

The IADL scale is a tool to assess independence in daily life among older adults residing in the community [4, 9]. The Korean version of IADL (K-IADL) was developed in consideration of cultural concerns and is widely used in Korea [10]. Commonly used IADL items contain concepts that are not measured equivalently by gender [11, 12], so it is necessary to be careful in interpreting the results. That is, IADL has the possibility of gender bias by overemphasizing tasks historically performed by women and overestimating dependency in men [9]. The limitations of some IADL items related to housework (e.g., cooking and laundry) in men are often caused by situational rather than functional challenges [13]. In other words, IADL performance according to gender might be over underestimated in some items because some tasks have not been experienced by some people due to the influence of lifelong gender roles. Considering that 91.2% of older adults in Korea are women performing most of the housework [14], gender differences in the ability to perform housework may affect the evaluation results of some IADL items. Therefore, it is necessary to identify the limitations in IADL items according to gender and change over time in older adults.

Factors related to IADL limitations in older adults have been reported as age, gender, living arrangement, marital status, education level, physical activity, social contact, quality of life, cognition, depression, chronic disease, trouble with pain, perceived health, fear of fall (FOF),

body mass index (BMI), and grip strength [8, 15–17]. These various factors influence deterioration of functional ability, including modifiable as well as non-modifiable factors. In particular, modifiable factors such as nutritional status, FOF, grip strength, and depression can be improved through a preventive approach [18], which is important for developing health policy in sociopolitical contexts.

Factors related to functional limitations of IADL are different for each item. Castellanos-Perilla and colleague report that limitations in preparing meals are associated with age, education, comorbidities, depression, and cognition, while only education is associated with limitations in use of medicine [8]. Factors associated with IADL limitations also differ according to gender. In a longitudinal study of older adults in China, social isolation for women and depression for men were associated with IADL limitations [19]. Among the IADL items, limitations in men were common in items related to housework, while limitations in women were related to other items [20]. Notably, in one previous study on the incidence rate of IADL limitations according to gender using big data from 23 countries, Korea was the only country with a higher incidence rate of IADL limitations in women than men [21]. It is necessary to identify factors associated with limitations in each IADL item according to gender in older Korean adults due to different cultural characteristics. Therefore, the purposes of this study are to explore changes in incidence of IADL limitations over 14 years and to identify factors associated with IADL limitations according to gender.

## Materials and method

### Sample

Secondary analysis was conducted using data from the Korean Longitudinal Study of Aging (KLoSA) [22]. Conducted in South Korea every two years from 2006 to the present, the KLoSA is an ongoing panel survey of personal and family information, health status, employment, income and consumption, subjective expectations, and quality of life. The original cohort was 10,254 randomly sampled adults over the age of 45 residing in the community across Korea. Sample retention rate in the most recent wave 8 was 77.1%. Herein, these eight waves representing 14 years of data (2006, 2008, 2010, 2012, 2014, 2016, 2018, and 2020) were analyzed to assess the changes and factors associated with IADL limitations in older adults. Inclusion criteria for the selected sample were 1) respondents aged 65 years and older at wave 1 (2006) showing, 2) full independence in IADL items at wave 1, and with complete data for all eight waves. The number of older adults included in the final analysis was 1,230 after excluding 9,024 from the baseline (n = 10,254) due to 1) death or not participating in–wave 8 (n = 4,537), 2) age younger than 65 years in 2006 (n = 4,116), 3) not participated during any wave 2–7 (n = 169) and 4) with at least one IADL limitation in 2006 (n = 202). The details of participant selection are shown in Fig 1.

### Ethical approval and consent to participate

The KLoSA has been approved by Statistics Korea (approval number: 336002) and was conducted with written consent from respondents based on the Declaration of Helsinki. All data are publicly available for academic research with de-identified information, rendering exempt ethical approval for this study.

### Measurement

**Dependent variable: Functional limitations.** Functional limitations were assessed using K-IADL [10]. The K-IADL consists of 10 activities of grooming, housework, preparing meals,

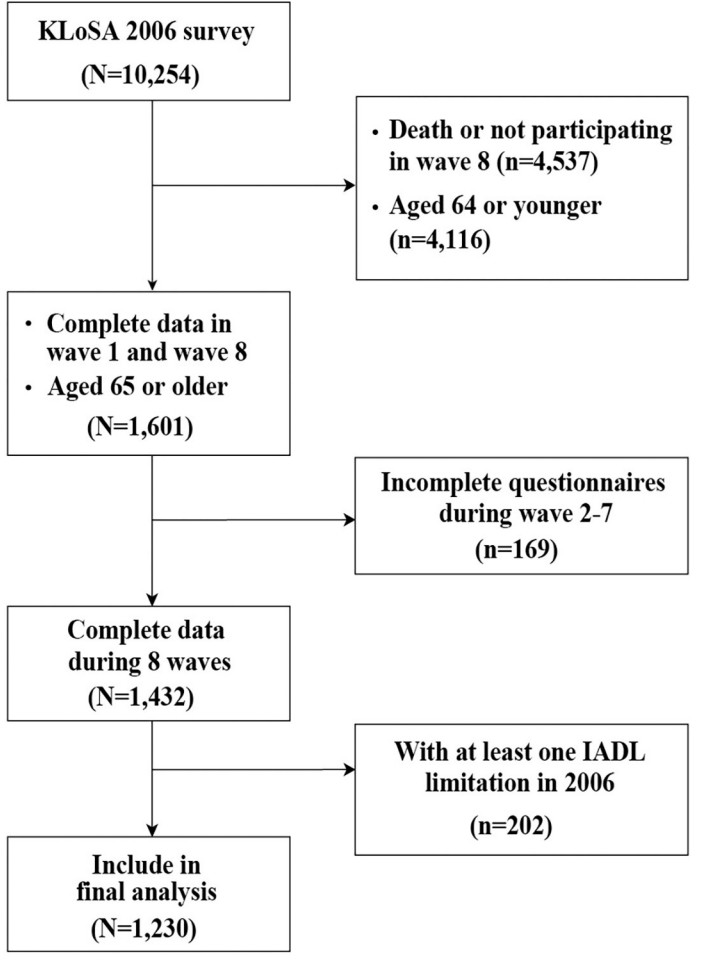

**Fig 1. Study flow chart.**

laundry, going out, using public transportation, shopping, handling money, using the telephone, and taking medicine. Scores for each item categorize respondents as independent (score = 0), partially dependent (score = 1), or fully dependent (score = 2) in that activity. In this study, an option for "do not perform" was not included in the responses. Total scores ranged from 0 to 20. In this study, IADL limitations were defined as a total IADL score of 1 or more [23]. For each item, determinations of partial and fully dependent were reclassified as IADL limitations.

**Independent variables.** Sociodemographic variables were age, living arrangement, marital status, education, and participation in social groups. Living arrangement was classified into living alone or living with others. Marital status was determined with the question "What is your current marital status?", with options of married or not being married. Participation in social groups was assessed by reclassifying respondents' answers to the question, "Are you participating in [any of] the following social group[s] (religious activity, volunteer work, and etc.)?" into a determination "yes" or "no".

Health-related factors included regular exercise, chronic disease, cognition, depression, trouble with pain, perceived health, perceived quality of life (QoL), experience of fall, FOF, grip strength, and BMI. Regular exercise was assessed by a "yes" or "no" response to the question, "Do you usually exercise more than once a week?" Pertinent chronic diseases were

hypertension, diabetes mellitus, cancer, pulmonary disease, liver disease, heart disease, cerebrovascular disease, psychiatric disease, and arthritis. The presence and number of chronic diseases in respondents was calculated by confirming any chronic disease diagnosed by a doctor, reclassified as "0" or "1 or more". Cognition was assessed using the Korean version of the Mini-Mental State Examination (K-MMSE) [24], which scores the following general cognitive functions: orientation to time and place, registration, attention and calculation, recall, language and visual construction. Total score ranges from 0–30, with higher scores indicating higher cognitive function. In this study, respondents' the total score were reclassified into dementia (0–17), mild cognitive impairment (18–23), or normal ($\geq$ 24) [24, 25]. Depression was classified as "yes" if participants had continuous depressive symptoms for the preceding two weeks or were taking an antidepressant and "no" if participants had no experience of depression. Trouble with pain was assessed with a single question; "Currently, do you have difficulty in performing daily activities due to pain?", with response options of "yes" or "no". Perceived health was questioned by asking, "What do you think of your health status?", with response options of "very good", "good", "fair", "poor", and "very poor". "Very good" and "good" were reclassified as "good", and the rest of the answers were reclassified as "poor". Perceived QoL was assessed with the question, "How satisfied are you with your overall quality of life in comparison to others of your age?", with participants asked to respond on a numeric rating scale of 0–10. Experience with falling was determined with the question, "Have you had a fall in the last two years?" with the answers either "yes" or "no", reflecting administration of the KLoSA every two years. Grip strength was calculated as the average value of both hands. FOF was measured with one question, "Are you usually afraid of falling?" Responses to the questions were reclassified from answers of "a little" and "very concerned" into "yes" and from "not at all concerned" into "no". BMI was calculated using body weight and height ($kg/m^2$). According to World Health Organization Asian Criteria of Obesity, BMI is classified into the following six categories: underweight (<18.5 $kg/m^2$); normal (18.5–22.9 $kg/m^2$); pre-obese (23.0–24.9 $kg/m^2$); obese class I (25.0–29.9 $kg/m^2$); obese class II (30.0–34.9 $kg/m^2$); and obese class III ($\geq$ 30 $kg/m^2$) [26]. In this study, BMI recategorized into non-obese (< 25 $kg/m^2$) and obese ($\geq$ 25 $kg/m^2$) based on the World Health Organization Asian Criteria of Obesity [27].

## Data analysis

Data analysis was conducted using statistical software IBM SPSS, version 23.0 (IBM Corp., Armonk, NY, USA). Descriptive analysis was performed to determine the mean (SD) for continuous variables and frequency (%) for non-continuous variables. Repeated measures analysis of variance (ANOVA) was applied to identify the changes in IADL score from wave 2 to wave 8 according to gender. Graphs related to the changes in the frequency of limitation occurrence during seven waves of analysis (wave 2 to wave 8) were expressed using Excel (Microsoft, Redmond, WA, USA). Univariate logistic regression was performed to identify gender difference in the factors associated with total IADL scores and the three IADL items showing the highest frequency of limitation in wave 8. Multivariate logistic regression analysis was performed with variables that were statistically significant in univariate analysis using the forward step. Statistical significance was accepted at $p < .05$. In present study, the sensitivity analysis was conducted to determine the influence of systemic selective attrition caused by follow-up loss during the data collection period on the factors of IADL limitations in older population. In the sensitivity analysis, 1,382 older adults were analyzed including people who incomplete questionnaires during wave 2 to 7 (S1 Fig). A univariate and multivariate logistic regression models were performed to check robustness of results in the present study. In addition, the results of sensitivity analysis are presented in the supplementary materials (S1 Fig, S1 Table).

## Results

A total of 1,230 older adults were included in our final analysis. The characteristics of participants at baseline (2006) are shown in Table 1. The mean age was 69.64 (±3.93) years (range 65–83) and 750 (61.0%) were female. Changes in IADL total scores over 14 years in total participants and in both men and women individually are presented in Fig 2. Total IADL scores increased over time, especially from wave 5, and changes were statistically significant in all groups [total participants: F = 104.82 ($p < .001$); men: F = 31.98 ($p < .001$); women: F = 74.36 ($p < .001$)]. Figs 3 and 4 shows the changes in the prevalence in the total and itemized IADL limitations over 14 years by total participants and by men and women individually. The prevalence of IADL limitations in all 10 items increased over time. The top three ranked items with high frequency at wave 8 were preparing meals (IADL 3), laundry (IADL 4), and using public transportation (IADL 6) for men and using public transportation (IADL 6), going out (IADL 5), and handling money (IADL 8) for women. Univariate analysis of total IADL scores and the top three ranked items for men and women are presented in Table 2. Although statistically significant predictors differed by group, the relatively common significant predictors were age, cognition, and grip strength. Factors associated with IADL limitations are presented in Table 3. The factors associated with total IADL limitations for total participants were age 80–89 years (OR: 2.87, 95% CI: 1.87–4.39), dementia (OR: 2.86, 95% CI: 1.74–4.71), grip strength (OR: 0.98, 95% CI: 0.96–1.00), and obesity (OR: 1.45, 95% CI: 1.04–2.01). On the other hand, the only associated factor for men was grip strength (OR: 0.95, 95% CI: 0.92–0.90). For women, age 80–89 years (OR: 3.25, 95% CI: 1.86–5.68), dementia (OR: 2.83, 95% CI: 1.63–4.92), FOF (OR: 1.56, 95% CI: 1.05–2.33), grip strength (OR: 0.93, 95% CI: 0.89–0.93), and obesity (OR: 1.65, 95% CI: 1.09–2.49) were significant.

Next, significant factors of each of the top three ranked IADL limitations differed by gender (Table 3). In men, age 80–89 years (OR: 2.06, 95% CI: 1.07–3.94) was associated with preparing meals; dementia (OR: 4.42, 95% CI: 1.08–18.07) with laundry; and dementia (OR: 4.96, 95% CI: 1.20–20.49) and grip strength (OR: 0.95, 95% CI: 0.91–0.99) with using public transportation. In women, associated factors for each item were as follows; age 80–89 years (OR: 3.55, 95% CI: 2.03–6.21), dementia (OR: 2.75, 95% CI: 1.57–4.83), grip strength (OR: 0.91, 95% CI: 0.86–0.95), and obesity (OR: 1.77, 95% CI: 1.16–2.71) for using public transportation; age 80–89 years (OR: 4.49, 95% CI: 2.53–7.95), dementia (OR: 2.54, 95% CI: 1.40–4.61), FOF (OR: 1.59, 95% CI: 1.02–2.47), grip strength (OR:0.90, 95% CI: 0.86–0.95), and obesity (OR: 1.93, 95% CI: 1.22–3.05) for going out; age 80–89 years (OR: 4.48, 95% CI: 2.52–7.98), dementia (OR: 2.96, 95% CI: 1.63–5.36), grip strength (OR:0.93, 95% CI: 0.88–0.98), and obesity (OR: 1.73, 95% CI: 1.12–2.78) for handling money.

The sensitivity analysis confirmed the result is similar to the main results of the present study. In univariate analysis, the relatively common significant factors in both men and women were age, cognition, trouble with pain, perceived health, FOF, and grip strength. In multivariate analysis, the significant factors of IADL limitations differed by gender. In men, age (OR: 1.93, 95% CI: 1.04–3.61), MCI (OR: 2.03, 95% CI: 1.14–3.59), and FOF (OR: 1.72, 95% CI: 1.13–2.60) were associated factors of IADL limitation. On the other hand, age (OR: 3.25, 95% CI: 1.98–5.33), dementia (OR: 2.35, 95% CI: 1.44–3.83), FOF (OR: 1.55, 95% CI: 1.04–2.33), and grip strength (OR: 0.95, 95% CI: 0.91–0.99) were associated factors of IADL limitation in older women. The result of the sensitivity analysis is presented in S1 Table.

## Discussion

This study identified the changes in IADL limitations and factors associated with IADL limitations according to gender in older adults in Korea. The IADL total score and the incidence of

**Table 1. Characteristics of the participants at baseline (2006) (N = 1,230).**

| Variables | n (%) | M± SD | range |
|---|---|---|---|
| Age (years) | | 69.64±3.93 | 65–83 |
| 65–75 | 1,105 (89.8) | | |
| >75 | 125 (10.2) | | |
| Sex | | | |
| Female | 750 (61.0) | | |
| Male | 480 (39.0) | | |
| Living arrangement | | | |
| Living alone | 169 (13.7) | | |
| Living with others | 1,061 (86.3) | | |
| Marital status | | | |
| Married | 901 (73.3) | | |
| Not married | 329 (26.7) | | |
| Education (years) | | | |
| 0–6 | 836 (68.0) | | |
| >7 | 394 (32.0) | | |
| Participate in social groups | | | |
| Yes | 833 (67.7) | | |
| No | 397 (32.3) | | |
| Regular exercise | | | |
| Yes | 462 (37.6) | | |
| No | 768 (62.4) | | |
| Number of chronic diseases | | 0.95±0.97 | 0–6 |
| $\geq 1$ | 756 (61.5) | | |
| 0 | 474 (38.5) | | |
| Cognition | | 24.62±4.67 | 0–30 |
| Dementia (0–17) | 104 (8.5) | | |
| MCI (18–23) | 285 (23.3) | | |
| Normal ($\geq 24$) | 829 (67.4) | | |
| Depression | | | |
| Yes | 153 (12.4) | | |
| No | 1,077 (87.6) | | |
| Trouble with pain | | | |
| Yes | 429 (34.9) | | |
| No | 801 (65.1) | | |
| Perceived health | | | |
| Good | 301 (24.5) | | |
| Bad | 929 (75.5) | | |
| Perceived QoL | | 61.44±20.53 | 0–100 |
| Falls (within 2 years) | | | |
| Yes | 61 (5.0) | | |
| No | 1,169 (95.0) | | |
| FOF | | | |
| Yes | 269 (21.9) | | |
| No | 961 (78.1) | | |
| Grip strength | | 22.74±7.55 | 1.25–46.50 |
| BMI | | 23.18±2.76 | 15.22–36.33 |
| Non-obesity | 915 (74.4) | | |

(*Continued*)

**Table 1.** (Continued)

| Variables | n (%) | M± SD | range |
|---|---|---|---|
| Obesity | 267 (21.7) | | |

Missing values: Cognition: (n = 12, 1.0%); grip strength (n = 69, 5.6%); BMI (n = 48, 3.9%); MCI: mild cognitive impairment; QoL: quality of life; FOF: fear of falling; BMI: body mass index

IADL limitations increased steadily for over 14 years in all participants, in agreement with previous studies [4, 8]. However, in the first six years (up to wave 4), there were no statistically significant differences in the changes in IADL total score for all participants nor in men and women independently and there was even a decrease in the incidence of IADL limitations in women. A three-year longitudinal study of Canadian older adults reports some improvement in IADL limitations across the three items of meal preparation, laundry, and telephone use [28]. Previous studies have reports that ready-made meals, use of home appliances, and improved accessibility can improve limitations in these IADL items [28, 29]. In other words, IADL limitations have been verifiably improved by changing situational and environmental factors to support, rather than undermine, the physical health of older adults [13]. A recent systematic review reported that multicomponent intervention (e.g., physical therapy, memory training, occupational therapy) are effective in improving IADL limitations in the short term (three months) as well as in the long term (12 months) [18]. Accordingly, it is necessary to establish active intervention plans including environmental modifications such as function-

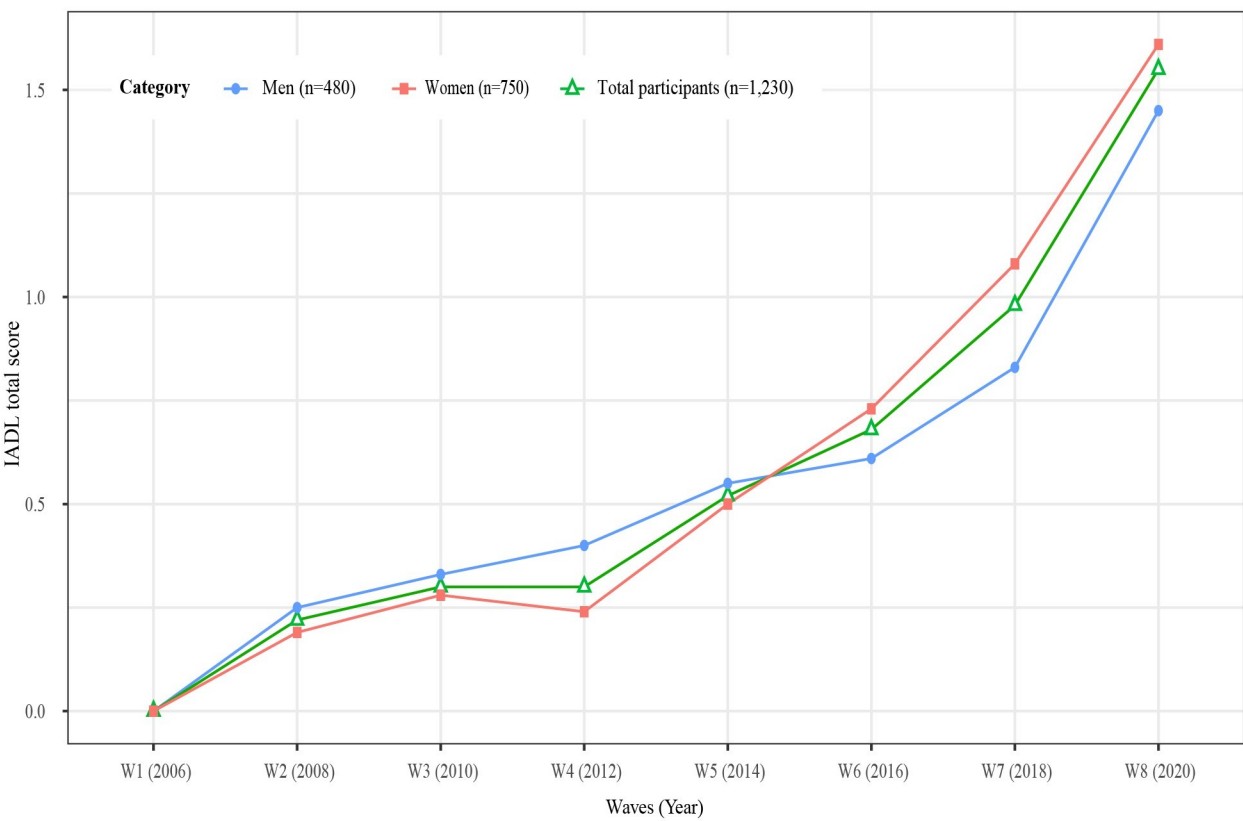

**Fig 2. Changes in IADL total scores over 14 years.**

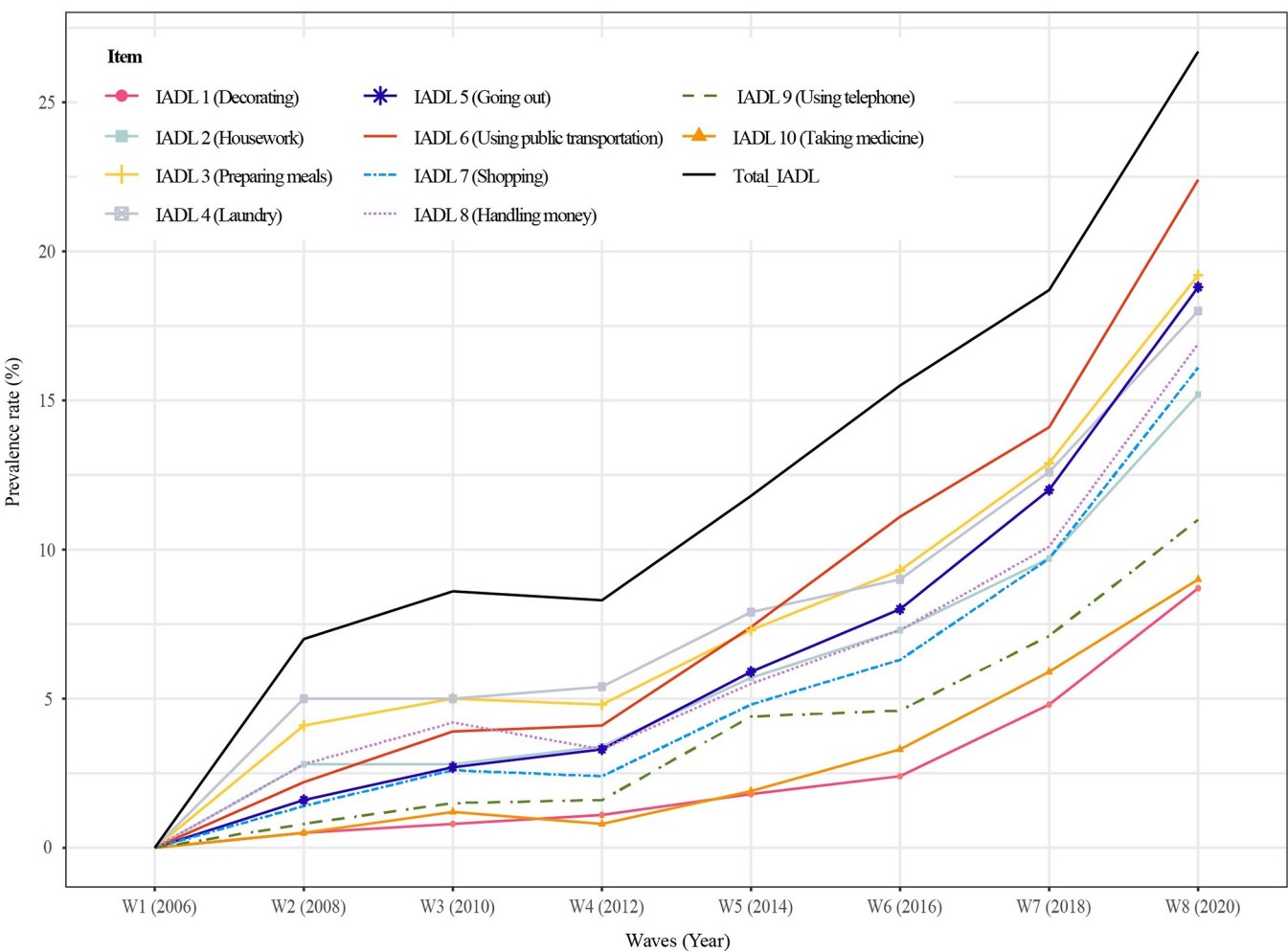

**Fig 3. Changes in IADL limitation prevalence over 14 years (Total participants).**

based housing to delay the onset of IADL limitations and to improve functional limitations as much as possible among older community- residing adults.

The top three ranked IADL items with a high incidence of limitations after 14 years in all participants were using public transportation, preparing meals, and going out. Previous research reports that the highest-incidences IADL limitations in the age group of 65 to 80years are housework and managing health care [4], which partially supports the findings of the present study. Here, the top three ranked IADL limitations differ according to gender. In men, the frequency of limitations was high in items related to housework such as preparing meals and laundry, whereas in women, the frequency of limitations was high in items going out and money management. This result differs from the results of a previous study conducted in the United States wherein the top ranked limitations in IADL activities did not differ by gender [20]. Although it is difficult to directly compare the two studies due to differences in age and data collection, the gender gap for IADL items (especially in preparing meals) in Korean older adults is meaningfully large. This likely reflect the perceptions of gender roles in Korea. According to a 2021 survey of family dynamics in Korea, 91.2% of married couples reported that wives do most of the housework [14]. In additional research in older adults aged 65 years and over, 97.4% of food-related work and 96.8% of laundry were performed by women [30].

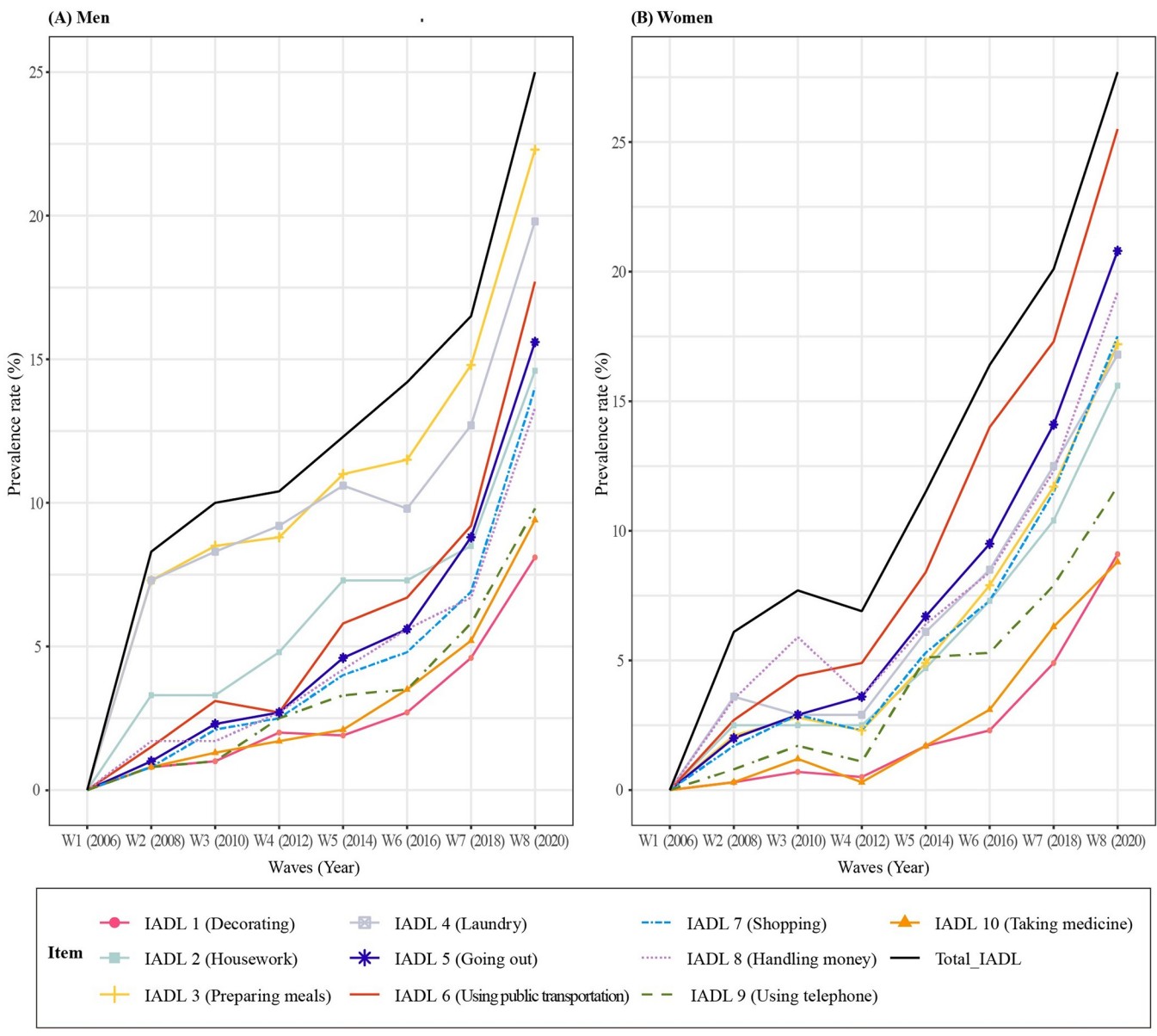

**Fig 4. Changes in IADL limitation prevalence over 14 years (Men, Women).**

As concluded by several studies as well as noted in this study, because most current Korean older men had no experience with household chores such as food preparation and laundry, caution should be taken in measuring gender-biased IADL items [12] and in interpreting results in IADL limitations. The KLoSA survey does not include a response option of "do not perform" for any IADL item for any survey respondent in Korea. Sheehan et al. [20] argue that a "do not perform" response to IADL items allows many different responses by gender and cohort, potentially causing systematic biases in IADL limitations and potentially influencing changes in IADL limitations according to gender [12, 20]. Further research must identify the factors influencing IADL limitations according to gender in Korean older adults using an IADL index that includes a response option of "do not perform" for participants with no experience with certain IADLs.

**Table 2. Univariate logistic regression models of total IADL limitations and the top three ranked IADL limitations after 14 years (N = 1,230).**

| Variables[†] | Total participants | Men | | | | Women | | | |
|---|---|---|---|---|---|---|---|---|---|
| | Total_IADL | Total_IADL | Preparing meals | Laundry | Using public transportation | Total_IADL | Using public transportation | Going out | Handling money |
| | OR (95% CI) | OR (95% CI) | OR (95% CI) | OR (95% CI) | OR (95% CI) | OR (95% CI) | OR (95% CI) | OR (95% CI) | OR (95% CI) |
| Age (years) | | | | | | | | | |
| 65–79 (ref) | | | | | | | | | |
| 80–89 | 3.47(2.38–5.06) ** | 2.09(1.11–3.94) * | 2.01(1.05–3.49) * | 1.49(0.74–3.01) | 1.53(0.74–3.15) | 4.65(2.87–7.54)** | 5.06(3.12–8.20) ** | 5.22(3.21–8.47) ** | 4.64(2.85–7.56) ** |
| Living arrangement | | | | | | | | | |
| Living alone | 0.83(5.70–1.22) | 0.45(0.10–1.03) | 0.53(0.12–2.38) | 0.60(0.14–2.77) | 0.30(0.04–0.50) | 0.82(0.55–1.12) | 0.79(0.52–1.21) | 0.73(0.46–1.16) | 0.69(0.42–1.12) |
| Living with others (ref) | | | | | | | | | |
| Marital status | | | | | | | | | |
| Married (ref) | | | | | | | | | |
| Not Married | 1.21(0.92–1.61) | 0.67(0.50–1.81) | 0.78(0.29–2.12) | 0.69(0.23–2.05) | 0.57(0.17–1.90) | 1.25(0.91–1.73) | 1.30(0.93–1.81) | 1.27(0.89–1.81) | 1.24(0.86–1.79) |
| Education (years) | | | | | | | | | |
| 0–6 | 1.40(1.06–1.85)* | 0.97(0.51–1.86) | 0.94(0.48–1.87) | 1.31(0.93–1.84) | 0.93(0.44–1.93) | 1.53(1.10–2.12) * | 1.50(1.07–2.09) * | 1.46(1.02–2.08) * | 1.69(1.17–2.45) * |
| More than 7 years (ref) | | | | | | | | | |
| Participate in social groups | | | | | | | | | |
| Yes (ref) | | | | | | | | | |
| No | 1.06(0.81–1.39) | 0.93(0.59–1.49) | 1.80(0.67–1.74) | 1.31(0.95–1.81) | 1.03(0.61–1.74) | 1.12(0.80–1.56) | 1.19(0.85–1.68) | 1.20(0.83–1.72) | 1.17(0.81–1.71) |
| Regular exercise | | | | | | | | | |
| Yes (ref) | | | | | | | | | |
| No | 1.12 (0.86–1.45) | 0.75 (0.50–1.13) | 0.70 (0.45–1.07) | 1.25 (0.90–1.74) | 0.86 (0.54–1.37) | 1.44 (1.01–2.06) * | 1.56 (1.08–2.26) * | 1.74 (1.16–0.63) ** | 1.75 (1.14–2.64) * |
| Number of chronic diseases | | | | | | | | | |
| 1+ | 1.08(0.83–1.40) | 1.17(0.77–1.77) | 1.45(0.90–2.33) | 1.45(0.92–2.29) | 1.45(0.90–2.33) | 1.0(0.71–1.40) | 1.11(0.78–1.57) | 1.18(0.81–1.72) | 1.02(0.69–1.49) |
| 0 (ref) | | | | | | | | | |
| Cognition | | | | | | | | | |
| Dementia | 4.14(2.72–6.29) ** | 3.34(0.80–13.6) | 3.85(0.94–15.7) | 4.42(1.08–18.07) * | 5.34(1.30–21.89) * | 4.29(2.70–6.82) ** | 4.09(2.56–6.52) ** | 3.60(2.23–5.83) ** | 3.75(2.30–6.12) ** |
| MCI | 1.33(0.98–1.81) | 1.85(1.3–3.35) * | 1.68(0.90–3.11) | 1.47(0.77–2.83) | 1.95(1.02–3.73) * | 1.24(0.85–1.80) | 1.42(0.97–2.08) | 1.70(0.84–1.92) | 1.32(0.86–2.03) |
| Normal (ref) | | | | | | | | | |
| Depression | | | | | | | | | |
| Yes | 1.21(0.84–1.76) | 1.43(0.70–2.93) | 1.47(0.70–3.07) | 1.29(0.59–2.81) | 2.18(0.98–4.86) | 1.11(0.72–1.72) | 1.04(0.66–1.64) | 1.01(0.62–1.64) | 0.81(0.48–1.37) |
| No (ref) | | | | | | | | | |
| Trouble with pain | | | | | | | | | |
| Yes | 1.37(1.06–1.78) * | 1.50(0.89–2.52) | 1.35(0.78–2.33) | 1.28(0.72–2.26) | 1.39(0.77–2.49) | 1.30(0.94–1.79) | 1.21(0.87–1.68) | 1.38(0.97–1.96) | 1.53(1.06–2.21) ** |
| No (ref) | | | | | | | | | |

*(Continued)*

**Table 2.** (Continued)

| Variables[†] | Total participants | Men | | | | Women | | | |
|---|---|---|---|---|---|---|---|---|---|
| | Total_IADL | Total_IADL | Preparing meals | Laundry | Using public transportation | Total_IADL | Using public transportation | Going out | Handling money |
| | OR (95% CI) | OR (95% CI) | OR (95% CI) | OR (95% CI) | OR (95% CI) | OR (95% CI) | OR (95% CI) | OR (95% CI) | OR (95% CI) |
| Perceived health | | | | | | | | | |
| Good (ref) | | | | | | | | | |
| Bad | 1.51(1.10–2.06) ** | 1.29(0.82–2.01) | 1.21(0.77–1.92) | 1.48(0.90–2.44) | 1.87(1.09–3.00) * | 1.71(1.08–2.69)* | 1.57(0.99–2.50) | 1.37(0.84–2.40) | 2.37(1.32–4.25) * |
| Perceived QoL | 0.99(0.98–0.99) * | 1.00(0.99–1.01) | 1.00(0.99(1.01) | 1.00(0.99–1.02) | 1.00(0.99–1.01) | | | | |
| Falls (within 2years) | | | | | | | | | |
| Yes | 0.890.49–1.62) | 1.35(0.41–4.45) | 1.05(0.28–3.88) | 0.73(0.16–3.36) | 1.41(0.38–5.23) | 0.76(0.35–1.52) | 0.76(0.37–1.55) | 0.99(0.98–1.00) * | 0.99(0.98–1.00) |
| No (ref) | | | | | | | | | |
| FOF | | | | | | | | | |
| Yes | 1.78(1.33–2.38) ** | 1.10(0.55–1.10) | 1.30(0.65–2.62) | 1.35(0.66–2.78) | 1.37(0.65–2.89) | 2.00(1.43–2.81) ** | 1.84(1.30–2.59) * | 1.89(1.31–2.72) * | 0.70(0.31–1.60) |
| No (ref) | | | | | | | | | |
| Grip strength | 0.97(0.95–0.98) ** | 0.95(0.91–0.99) ** | 0.96(0.92–0.99) * | 0.96(0.90–1.00) | 0.94(0.90–0.98) ** | 0.91(0.87–0.95)** | 0.89(0.85–0.93) ** | 0.89(0.85–0.93) ** | 1.72(1.18–2.52) ** |
| BMI | | | | | | | | | |
| Non-obesity (ref) | | | | | | | | | |
| Obesity | 1.05(1.00–1.10) * | 0.99(0.58–0.97) | 0.94(0.53–1.66) | 0.97(0.54–1.75) | 0.95(0.52–1.76) | 1.50(1.04–2.17) * | 1.50(1.03–2.19) * | 1.07(1.01–1.15) * | 1.51(1.00–2.60) * |

OR: odds ratio; CI: confidence interval; IADL: instrumental activities of daily living; MCI: mild cognitive impairment; QoL: quality of life; FOF: fear of falling; BMI: body mass index

*p<0.05

**p<0.01, † Data of baseline (2006)

Unlike men, older women in the present study had a high frequency of limitations in mobility-related items of IADL. This finding might result from the earlier loss of lower extremity muscle strength in women than in men due to the effects of hormonal changes (i.e., reduction of estrogen) mainly because of menopause, along with a decrease in muscle mass [31]. A decrease in lower extremity muscle strength leads to decreases in physical performance abilities such as gait speed and balance ability, which act as a functional disadvantage in women [32]. Lower extremity muscle strength and physical performance ability in women are related to a decline in mobility, which can decrease older women's ability to use public transportation and to go out. To prevent these IADL limitation in older women, active intervention is needed to amplify the range of function in individuals by providing an interventional or assistive device to improves muscle strength [33, 34].

In this study, dementia, grip strength, and BMI are associated with total IADL limitations in all participants, supporting the findings of previous studies [8, 27, 35]. On the other hand, differences are observed in the factor associated with total IADL limitations according to gender. Although only grip strength was related to total IADL limitations in men; age, dementia, FOF, grip strength, and BMI were identified in women. These findings are different from a

**Table 3. Multivariate logistic regression models of total IADL limitations and the top three ranked IADL limitations after 14 years (N = 1,230).**

| Variables† | Total participant | Men | | | | Women | | | |
|---|---|---|---|---|---|---|---|---|---|
| | Total_IADL | Total_IADL | Preparing meals | Laundry | Using public transportation | Total_IADL | Using public transportation | Going out | Handling money |
| | OR (95% CI) | OR (95% CI) | OR (95% CI) | OR (95% CI) | OR (95% CI) | OR (95% CI) | OR (95% CI) | OR (95% CI) | OR (95% CI) |
| Age (years) | | | | | | | | | |
| 65–79 (ref) | | | | | | | | | |
| 80–89 | 2.87(1.87–4.39) ** | | 2.06(1.07–3.94) * | | | 3.25(1.86–5.68) ** | 3.55(2.03–6.21) ** | 4.49(2.53–7.95) ** | 4.48(2.52–7.98) ** |
| Cognition | | | | | | | | | |
| Dementia | 2.86(1.74–4.71) ** | | | 4.42(1.08–18.07) * | 4.96(1.20–20.49) * | 2.83(1.63–4.92) ** | 2.75(1.57–4.83) ** | 2.54(1.40–4.61) ** | 2.96(1.63–5.36) ** |
| MCI | 1.14(0.80–1.63) | | | 1.47(0.77–0.83) | 1.51(0.75–3.04) | 1.08(0.70–1.65) | 1.22(0.78–1.90) | 1.22(0.75–1.97) | 1.35(0.83–2.18) |
| Normal (ref) | | | | | | | | | |
| FOF | | | | | | | | | |
| Yes | | | | | | 1.56(1.05–2.33) * | | 1.59(1.02–2.47) * | |
| No (ref) | | | | | | | | | |
| Grip strength | 0.98(0.96–1.00) * | 0.95(0.92–0.99) * | | | 0.95(0.91–0.99) * | 0.93(0.89–0.98) * | 0.91(0.86–0.95) ** | 0.90(0.86–0.95) ** | 0.93(0.88–0.98) * |
| BMI | | | | | | | | | |
| Non-obesity (ref) | | | | | | | | | |
| Obesity | 1.45(1.04–2.01) * | | | | | 1.65(1.09–2.49) * | 1.77(1.16–2.71) * | 1.93(1.22–3.05) * | 1.73(1.12–2.78) * |

OR: odds ratio; CI: confidence interval; IADL: instrumental activities of daily living; MCI: mild cognitive impairment; FOF: fear of falling; BMI: body mass index

*$p<0.05$

**$p<0.01$

† Data of baseline (2006)

previous study in which factors related to IADL limitations were similar in men and women [19]. In addition, in the current study, factors associated with the top three ranked IADL items differ according to item and gender. These results may simultaneously reflect the characteristics of Korean culture and gender roles. Therefore, customized approaches are necessary to prevent IADL limitations in Korean older adults by considering the factors that reflect these characteristics.

Grip strength is an associated factor of IADL limitations in both male and female older adults, which is similar to a study that reports that grip strength in older adults had a predictive association with IADL limitations. That research explains that a decrease in grip strength caused difficulty in shopping, preparing meals, and performing general household chores [35]. Because muscle weakness is closely related to personal/social care needs as well as decline in daily QoL, active interventions to improve muscle strength in both men and women are needed.

Cognitive function is a predictive factor in laundry and use of public transportation in men and in all IADL items in women, which is consistent with the findings of previous studies [28, 36]. Instrumental activities of daily living are affected by cognitive decline due to the complex and multifaceted nature of cognition [36]. In particular, Passler et al. report that cognitive

decline is the most powerful predictor of IADL limitations, increasing the incidence of IADL limitations at 10-year follow-up by an odds ratio of 1.48 times [36]. Cognitive decline and IADL are mutual influencing factors, and older adults with cognitive decline are more likely to develop IADL limitations [37]. In older adults with mild cognitive impairment, loss of independence in IADL activities is a major predictor for future dementia [38]. Accordingly, because functional dependence is accelerated by cognitive impairment and is a major predictor of dementia [37], it is necessary to periodically assess cognitive function and carefully examine the relationship between the two factors in older adults. In addition, with regard to IADL items related to cognitive function, it is necessary to provide individual customized services according to gender.

Obesity was consistently shown to be an associated predictor of IADL limitations in female older adults, which is supported by the findings of previous studies [19, 27]. In particular, Chiu et al. report that BMI had an effect on IADL limitations in women but not in men [19]. More recently, however, it has been reported that the risk of IADL limitations increases when there is sarcopenia or sarcopenic obesity rather than obesity alone [39]. Sarcopenic obesity is caused by age-related changes in muscle mass, which are closely related to functional limitations [40]. Therefore, BMI control is needed to prevent and delay IADL limitations in women, and individual-level intervention to maintain muscle mass are required.

FOF is a predictor of IADL limitations only in women, consistent with the results of previous studies [41, 42]. FOF increases the proportion of IADL limitations among older adults [41]; in particular, FOF has been identified as a major associated factor (OR: 3.43, 95% CI 2.15–5.47) causing functional decline in older women [42]. FOF in older adults results in decreased physical function along with restricting activities, decreased social activity, and decline in cognitive function [43]. In particular, based on the results of this and previous studies, FOF influences the occurrence of IADL limitations not only in the short-term (two years), but also in the long-term (14 years) in older women. Because FOF is a potentially reversible factor, strategies for detection and personalized intervention are essential to prevent limitations together with multiple negative consequences [41].

## Strength and limitations

The present study has some limitations. First, the original data were collected with self-reported questionnaire. Self-reporting is an incomplete indicator of actual behavior and may cause potential biases in the results. Second, because this study uses secondary data, certain associated factors of IADL limitations reported in previous studies (e.g., economic status, performance of activities, frailty, walking speed, balance) were not included. Third, associated factors of the top three ranked items with the highest frequency in the K-IADL were identified. It is necessary to additionally verify factors for other IADL items to ensure a more robust individualized approach to prevention of IADL limitations. Despite several limitations, this study has the following strengths. First, the results of sensitivity analyses were similar to the main results, and it was proven that the influence of systematic selective attrition was not significant, confirming the robustness of the findings. Second, in community-residing older adults, changes in IADL limitations over 14 years were confirmed through the national data, and these changes varied by gender. There was difference in items with a high frequency of limitations according to gender, which is a reflection of the cultural characteristics in Korean subjects. In this study, different associated predictors were observed according to total participants, gender, and IADL items. These findings suggests that tailored interventions that reflects the characteristics of each older adult are required to prevent limitations and improve function in community-residing older adults.

## Conclusion

Total IADL scores in this study increased significantly after 14 years. Because this outcome can be improved through changes in situational factors and active interventions, robust policy approaches to delay the occurrence of limitations are needed. Differences are shown in the major items of IADL limitations according to gender and in associated predictors. If an approach is based on specific predictors of overall IADL limitations of older Korean adults, then limitations can be minimized by tailoring the interventions according to gender and individual factors. In particular, in a situation where perceptions of gender roles are biased, such as in Korea, functional limitation determination using the IADL may not fully reflect the characteristics of both genders. To effectively improve well-being in older adults, data should be analyzed by gender.

## Supporting information

**S1 Fig. Flow chart for sensitivity analysis.** The sensitivity analysis was conducted to determine the influence of systemic selective attrition caused by follow-up loss during the data collection period on the factors of IADL limitations in older population. In the sensitivity analysis, 1,382 older adults were analyzed including people who incomplete questionnaires during wave 2 to 7.
(TIF)

**S1 Table. Univariate and multivariate logistic regression of total IADL limitations after 14 years including older adults not participated during wave 2–7 (N = 1,382).**
(DOCX)

## Author Contributions

**Conceptualization:** SeolHwa Moon, Eunmi Oh, Gwi-Ryung Son Hong.

**Data curation:** SeolHwa Moon.

**Formal analysis:** SeolHwa Moon.

**Funding acquisition:** Gwi-Ryung Son Hong.

**Methodology:** SeolHwa Moon, Eunmi Oh.

**Project administration:** Gwi-Ryung Son Hong.

**Supervision:** Gwi-Ryung Son Hong.

**Visualization:** Daum Chung.

**Writing – original draft:** SeolHwa Moon, Eunmi Oh, Daum Chung, Gwi-Ryung Son Hong.

**Writing – review & editing:** SeolHwa Moon, Eunmi Oh, Daum Chung, Gwi-Ryung Son Hong.

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
