## [Decision Letter · Decision Letter 0]

12 Jul 2023

PONE-D-22-33705Changes in Instrumental Activities Daily Living limitations and their associated factors according to gender in community-residing older adults: A longitudinal cohort studyPLOS ONE

Dear Dr. Hong,

Thank you for submitting your manuscript to PLOS ONE. After careful consideration, we feel that it has merit but does not fully meet PLOS ONE’s publication criteria as it currently stands. Therefore, we invite you to submit a revised version of the manuscript that addresses the points raised during the review process. The reviewers have made major and minor suggestions to improve your manuscript. I invite you to respond to their comments and revise your manuscript accordingly. The comments of the reviewer(s) are included at the bottom of this letter.

We look forward to receiving your revised manuscript.

Kind regards,

Fabíola Bof de Andrade

Academic Editor

PLOS ONE

Journal Requirements:

“This research was supported by the Translational Research Program for Care Robots funded by the Ministry of Health & Welfare, Republic of Korea (grant number: HK21C0008010021).

“This research was supported by the Translational Research Program for Care Robots funded by the Ministry of Health & Welfare, Republic of Korea (grant number: HK21C0008010021). “

“Funding acquisition: Gwi-Ryung Son Hong

Grant numbers: HK21C0008010021

This research was supported by the Translational Research Program for Care Robots funded by the Ministry of Health & Welfare, Republic of Korea (grant number: HK21C0008010021).

Please include your amended statements within your cover letter; we will change the online submission form on your behalf.’

Additional Editor Comments (if provided):

The reviewers have made suggestions to improve your manuscript. Therefore, I invite you to respond to their comments and revise your manuscript accordingly. The comments of the reviewer(s) are included at the bottom of this letter.

Reviewers' comments:

Reviewer's Responses to Questions

**Comments to the Author**

1. Is the manuscript technically sound, and do the data support the conclusions?

Reviewer #1: Yes

Reviewer #2: Yes

2. Has the statistical analysis been performed appropriately and rigorously? 

Reviewer #1: Yes

Reviewer #2: Yes

3. Have the authors made all data underlying the findings in their manuscript fully available?

Reviewer #1: No

Reviewer #2: Yes

4. Is the manuscript presented in an intelligible fashion and written in standard English?

Reviewer #1: Yes

Reviewer #2: Yes

5. Review Comments to the Author

Reviewer #1: Thank you for the opportunity to review the manuscript entitled: “Changes in Instrumental Activities Daily Living limitations and their associated factors 2 according to gender in community-residing older adults: A longitudinal cohort study” The manuscript is well written and analyzed. I offer minor comments below in the spirit of improving the submission.

The authors analyze only respondents who appeared in waves from 2004-2018. Was there systematic selective attrition? If I am understanding this correctly, about 23% of respondents were excluded? Was any sensitivity analyses conducted on who this group was? Could imputation be used for respondents with missing data? I think the mortality selection could be leading to the decrease finding for women in the first few waves.

In the Health and Retirement Study, IADLs also have an option for “Do not do,” is that available here to analyze as well?

Should the tables be labeled as logistic models if they are predicting the sum? Sorry if I am not understanding this correctly.

Could table 2 be a figure instead? Or the figures included at the bottom replace it?

The figures are difficult to see clearly, could they be made larger?

I could not tell if the data are available, could that be made more clear?

Reviewer #2: The manuscript addresses an important topic from a perspective that there are few studies exploring the changes in incidence of IADL limitations over a long period as well as identifying the related factors of IADL limitations according to gender. The authors use data from a broad and recognized population-based survey with a well know method. However, there are still some gaps that need to be addressed in order for the text to be suitable for publication.

Data Availability: please detail which restrictions would apply.

ABSTRACT

- Conclusion: please insert a statement regard the changes in incidence of IADL limitations over the years.

INTRODUCTION

- It would be advisable to include a discussion of changes in the incidence/prevalence of IADL limitations over time according to the literature.

MATERIALS AND METHODS

- Independent variables:

1) Please justify why to classify “regular exercise” as a sociodemographic variable.

2) Which is the category that includes the answer “1” for the “number of chronic diseases” variable? The description provided in that section does not correlates to Table 1 categories.

3) Please justify the reason for using the 2-year cutoff point in relation to the variable "experience of fall".

RESULTS

- I suggest to standardize the period in which the changes in IADL is assessed. In the abstract, the authors referred to “14 years” whereas in the Results, it appears a period of 12 years of changes. It seems there is an ambiguous interpretation of the period including or not including baseline wave (2006).

- Please, cite the Table source of the data in the last paragraph of the Results section.

- Table 2: please, insert subtitles for this Table. It would be advisable to include a column with the baseline records (2006) for the IADL total scores.

- Tables 3 and 4: For a better visualization, I suggest a division between the columns containing data of the Total of the sample; Men; and Women. Furthermore, it is necessary to inform the year/wave of the data in Table titles.

6. PLOS authors have the option to publish the peer review history of their article (what does this mean?). If published, this will include your full peer review and any attached files.

Reviewer #1: No

Reviewer #2: **Yes: **Eduardo José Pereira Oliveira

---

## [Author Response · Author response to Decision Letter 0]

23 Aug 2023

Thank you for your comments! We have revised the manuscript based on the reviewer’s comments, please see the page number for each reviewers’ comment. We also got the professional editing service throughout the manuscript.

Reviewer #1 

1. The authors analyze only respondents who appeared in waves from 2004-2018. Was there systematic selective attrition? If I am understanding this correctly, about 23% of respondents were excluded? Was any sensitivity analyses conducted on who this group was? Could imputation be used for respondents with missing data? I think the mortality selection could be leading to the decrease finding for women in the first few waves.

→Thank you for your comment! Unfortunately, we did not conduct sensitivity analysis in this study. We described this issue in limitation about careful interpretation of the results of this study (on page 22: Finally, systematic selective attrition caused by death and loss follow-up during the data collection period was not considered. In other words, there may be possibility of attrition bias in this study that does not reflect the characteristics of vulnerable group with the greatest limitation of IADL, so it is necessary to be careful in interpreting the results. In further study, it is suggested to conduct sensitivity analysis to minimize the attrition bias of the influencing factors of IADL limitation occurrence in this population.)

2. In the Health and Retirement Study, IADLs also have an option for “Do not do,” is that available here to analyze as well?

→ Thank you for your great comment! The IADL in KLoSA survey does not have an option ‘do not do’ response. We added the issue regarding on the response format in discussion (on page 18: The KLoSA survey does not include a response option of “do not perform” for any IADL item for any survey respondent in Korea. Sheehan et al. [20] argue that a “do not perform” response to IADL items allows many different responses by gender and cohort, potentially causing systematic biases in IADL limitations and potentially influencing changes in IADL limitations according to gender [12, 20]. Further research must identify the factors influencing IADL limitations according to gender in Korean older adults using an IADL index that includes a response option of “do not perform” for participants with no experience with certain IADLs.) 

3. Should the tables be labeled as logistic models if they are predicting the sum? Sorry if I am not understanding this correctly.

→ Sorry for the confusion! We tried to added the longitudinal line in tables 2 and 3 to improve the readers’ better understanding the study result. 

4. Could table 2 be a figure instead? Or the figures included at the bottom replace it?

→ Yes, we agreed on your suggestion, and changed Table 2 to figure. We revised the section in main text (on page 10: Total IADL scores increased over time, especially from wave 5, and changes were statistically significant in all groups [total participants: F = 104.82 (p < .001); men: F = 31.98 (p < .001); women: F = 74.36 (p < .001)]) and Please see Figure 2.

5. The figures are difficult to see clearly, could they be made larger?

→ Yes, we improved the resolution and size of the figures. Please see Figures 3 and 4.

6. I could not tell if the data are available, could that be made more clear?

→ Sorry for the confusion. Since we used public data for this study, we changed data availability to ‘all data are fully available without restriction’.

Reviewer #2 Comment

The manuscript addresses an important topic from a perspective that there are few studies exploring the changes in incidence of IADL limitations over a long period as well as identifying the related factors of IADL limitations according to gender. The authors use data from a broad and recognized population-based survey with a well know method. However, there are still some gaps that need to be addressed in order for the text to be suitable for publication.

→ Thank you for your comments!

1. Data Availability: please detail which restrictions would apply

→ Sorry for the confusion. Since we used public data for this study, we changed data availability to ‘all data are fully available without restriction’.

2. ABSTRACT 

Conclusion: please insert a statement regard the changes in incidence of IADL limitations over the years.

→ Thank you for your comment! We added the statement in conclusion of the abstract(The incidence of IADL limitations gradually increased in all participants over a 14-year period).

3. INTRODUCTION

It would be advisable to include a discussion of changes in the incidence/prevalence of IADL limitations over time according to the literature.

→ regarding changes in the incidence/prevalence of IADL limitations over time using two literatures (on page 4: Limitations in IADL in older adults increase with age, with previous research reporting gradual increase over a 10-year period [4], as well as in a study with just three years of follow-up [8]. Most previous research examines changes in the incidence of IADL limitations in all segments of older populations together. Because incidence and limitations in IADL vary based on gender for different activities, it is important to assess these changes by gender to understand functional limitations among older adults).

4. MATERIALS AND METHODS

- Independent variables:

1) Please justify why to classify “regular exercise” as a sociodemographic variable.

→ We classified “regular exercise” as health-related factors, and updated the method section (on page 8) and Tables 1, 2. 

2) Which is the category that includes the answer “1” for the “number of chronic diseases” variable? The description provided in that section does not correlates to Table 1 categories.

→ We calculated the number of chronic diseases (included hypertension, diabetes mellitus, cancer, pulmonary disease, liver disease, heart disease, cerebrovascular disease, psychiatric disease, and arthritis) by checking the chronic disease diagnosed by the doctor, and updated the method section (on page 8).

3) Please justify the reason for using the 2-year cutoff point in relation to the variable "experience of fall".

→ KLoSA survey were conducted every two years, and “fall experience” were asked for the experience during the past 2 years. We added the statement in methods section (on page 9: Experience with falling was determined with the question, “Have you had a fall in the last two years?”, with answers of either “yes” or “no,” reflecting administration of the KLoSA every two years).

5. RESULTS

1) I suggest to standardize the period in which the changes in IADL is assessed. In the abstract, the authors referred to “14 years” whereas in the Results, it appears a period of 12 years of changes. It seems there is an ambiguous interpretation of the period including or not including baseline wave (2006).

→ Thank your comment! We revised the observation period to 14 years based on baseline (2006, wave 1) throughout the main text. 

2) Please, cite the Table source of the data in the last paragraph of the Results section.

→ Thank your comment! We added table source (Table 3) (on page on 11).

3) Table 2: please, insert subtitles for this Table. It would be advisable to include a column with the baseline records (2006) for the IADL total scores.

→ Thank your comment! We changed Table 2 to Figure 2 including the IADL total score of baseline (2006) in Figure 2-4.

4) Tables 3 and 4: For a better visualization, I suggest a division between the columns containing data of the Total of the sample; Men; and Women. Furthermore, it is necessary to inform the year/wave of the data in Table titles.

→ Thank your comment! We revised Table 2 and 3 (page on 14-16).

---

## [Decision Letter · Decision Letter 1]

8 Oct 2023

PONE-D-22-33705R1Changes in Instrumental activities daily living limitations and their associated factors according to gender in community-residing older adults: A longitudinal cohort studyPLOS ONE

Dear Dr. Hong,

Thank you for submitting your manuscript to PLOS ONE. After careful consideration, we feel that it has merit but does not fully meet PLOS ONE’s publication criteria as it currently stands. Therefore, we invite you to submit a revised version of the manuscript that addresses the points raised during the review process.

The two original reviewers have read the revised paper and signaled their satisfaction with the authors' responses.  However each raises one additional point that they would like to see addressed.  Reviewer 1 requests an attrition analysis so that there is more information about loss from the sample.  Reviewer 2 requests addition attention to the classification of obesity; as I interpret it the reviewer would simply like a language change, and not the creation of an additional category (dividing >=25 into overweight and obese).  Please just change the name of the category so that it is clear that not everyone over BMI 25 is considered obese.  Unless additional issues arise, I should not need to send the paper out for review again, but can make the determination myself.

We look forward to receiving your revised manuscript.

Kind regards,

Ellen L. Idler

Academic Editor

PLOS ONE

Journal Requirements:

Reviewers' comments:

Reviewer's Responses to Questions

**Comments to the Author**

1. If the authors have adequately addressed your comments raised in a previous round of review and you feel that this manuscript is now acceptable for publication, you may indicate that here to bypass the “Comments to the Author” section, enter your conflict of interest statement in the “Confidential to Editor” section, and submit your "Accept" recommendation.

Reviewer #1: All comments have been addressed

Reviewer #2: All comments have been addressed

2. Is the manuscript technically sound, and do the data support the conclusions?

Reviewer #1: Yes

Reviewer #2: Yes

3. Has the statistical analysis been performed appropriately and rigorously? 

Reviewer #1: I Don't Know

Reviewer #2: Yes

4. Have the authors made all data underlying the findings in their manuscript fully available?

Reviewer #1: Yes

Reviewer #2: Yes

5. Is the manuscript presented in an intelligible fashion and written in standard English?

Reviewer #1: Yes

Reviewer #2: Yes

6. Review Comments to the Author

Reviewer #1: I think an attrition analysis would improve the analyses and results. This would benefit the results substantially and you could run a simple logistic model to do it.

Reviewer #2: Thanks for review the second version of this important manuscript which addresses an important topic from a perspective that there are few studies exploring the changes in incidence of IADL limitations over a long period as well as identifying the related factors of IADL limitations according to gender. I thank and congratulate the authors for considering my recommendations and amending the text accordingly. Finally, I would like to suggest the following change:

- METHODS: Please, revise the “obese (≥25kg/m2)” classification in BMI variable. Overweight/obese category might be considered.

7. PLOS authors have the option to publish the peer review history of their article (what does this mean?). If published, this will include your full peer review and any attached files.

Reviewer #1: No

Reviewer #2: **Yes: **Eduardo José Pereira Oliveira

---

## [Author Response · Author response to Decision Letter 1]

13 Nov 2023

Thank you for your comments! We have revised the manuscript based on the reviewer’s comments.

---

## [Editor Report · Decision Letter 2]

19 Nov 2023

PONE-D-22-33705R2Changes in Instrumental activities daily living limitations and their associated factors according to gender in community-residing older adults: A longitudinal cohort studyPLOS ONE

Dear Dr. Hong,

Thank you for submitting your manuscript to PLOS ONE. After careful consideration, we feel that it has merit but does not fully meet PLOS ONE’s publication criteria as it currently stands. Therefore, we invite you to submit a revised version of the manuscript that addresses the points raised during the review process.

I have re-read the manuscript submitted as Revision 2.  The previous reviewers had already advised that we accept the manuscript, but each made a small suggestion, which I included in the previous editor's decision, so that the acceptance was conditional on addressing them.  The first suggestion was that the authors perform a sensitivity analysis to better understand the attrition from the sample.  Although the authors have added a table to the supplemental files, the results are not described in the text, and the table title is unclear.  The N for the table is 1382.   The N for the analysis is 1230.  The Supplemental table's title suggests that it is an analysis of data from those missing from the final sample because of missing data in waves 2-7, but the language is unclear. Moreover, the added text describing the results of the sensitivity analysis do not refer to attrition at all.  Please explain the data sources and purpose of the analysis.The second suggestion was to change the language for the covariate of obesity.  It seems that no change was made, only a reference to the World Health Organization's categorization was made.  Please explain your decision.

We look forward to receiving your revised manuscript.

Kind regards,

Ellen L. Idler

Academic Editor

PLOS ONE
---

## [Author Response · Author response to Decision Letter 2]

14 Dec 2023

Thank you for your comments! We have revised the manuscript based on the editor’s comments.

The details are included in the file of reviewer's comment.

---

## [Editor Report · Decision Letter 3]

19 Dec 2023

Changes in Instrumental activities daily living limitations and their associated factors according to gender in community-residing older adults: A longitudinal cohort study

PONE-D-22-33705R3

Dear Dr. Hong,

We’re pleased to inform you that your manuscript has been judged scientifically suitable for publication and will be formally accepted for publication once it meets all outstanding technical requirements.

Kind regards,

Ellen L. Idler

Academic Editor

PLOS ONE
---

## [Editor Report · Acceptance letter]

3 Jan 2024

PONE-D-22-33705R3 

PLOS ONE

Dear Dr. Hong, 

I'm pleased to inform you that your manuscript has been deemed suitable for publication in PLOS ONE. Congratulations! Your manuscript is now being handed over to our production team.

Kind regards, 

on behalf of

Professor Ellen L. Idler 

Academic Editor

PLOS ONE